# Pharmacokinetics of Tildipirosin in Ewes after Intravenous, Intramuscular and Subcutaneous Administration

**DOI:** 10.3390/ani10081332

**Published:** 2020-08-01

**Authors:** Juan Sebastián Galecio, Elisa Escudero, José Joaquín Cerón, Giuseppe Crescenzo, Pedro Marín

**Affiliations:** 1Department of Pharmacology, Faculty of Veterinary Medicine, University of Murcia, 30100 Murcia, Spain; escudero@um.es; 2Escuela de Medicina Veterinaria, Colegio de Ciencias de la Salud, Universidad San Francisco de Quito, Cumbayá EC 170157, Ecuador; 3Interdisciplinary Laboratory of Clinical Pathology, Interlab–UMU, University of Murcia, 30100 Murcia, Spain; jjceron@um.es; 4Department of Veterinary Medicine, University of Bari Aldo Moro, 70010 Valenzano, BA, Italy; giuseppe.crescenzo@uniba.it

**Keywords:** macrolides, pharmacokinetics, sheep, ewes, tildipirosin

## Abstract

**Simple Summary:**

Pneumonia is a significant cause of death in sheep flocks. Thus, antibiotics are essential for the treatment of bacterial pneumonia to reduce morbidity and mortality, but few drugs are specifically labeled for clinical use in sheep. Many of the antimicrobial clinical prescriptions that occur in sheep are classified as extra-label use, with dosage, administration frequency, indications, and drug withdrawal times usually being extrapolated from information reported in other species. Thus, the objective of this study was to determine the disposition kinetics of tildipirosin after intravenous, intramuscular, and subcutaneous administration in sheep. Throughout the experiment, all ewes were healthy and no adverse reactions were recorded. The apparent volume of distribution was high, indicating a wide distribution in the body, which can be attributed to a significant fraction of tildipirosin inside the cells, and its expected activity against intracellular bacteria. Following intramuscular administration, tildipirosin was rapidly absorbed even to a greater extent when compared to subcutaneous administration. Most of the adsorbed tildipirosin after intramuscular and subcutaneous administrations were available in the body (>70%). In brief, the excellent tolerability of this formulation and the suitable disposition of tildipirosin in the body makes it an alternative for sheep use, in conditions where the administration of antibiotics is needed to observe desired effects with the benefits of scant manipulation of animals.

**Abstract:**

A single-dose disposition kinetics for tildipirosin was evaluated in clinically healthy ewes (*n* = 6) after intravenous (IV), intramuscular (IM), and subcutaneous (SC) administration of a commercial formulation. Tildipirosin concentrations were determined by high-performance liquid chromatography with ultraviolet detection. Plasma concentration-time data was calculated by non-compartmental pharmacokinetic methods. The apparent volume of distribution (Vz) of tildipirosin after IV administration was 5.36 ± 0.57 L/kg suggesting a wide distribution in tissues and inside the cells. The elimination half-life (t½λz) was 17.16 ± 2.25, 23.90 ± 6.99 and 43.19 ± 5.17 h after IV, IM and SC administration, respectively. Following IM administration, tildipirosin was rapidly absorbed (tmax = 0.62 ± 0.10 h) even to a greater extent than after SC administration. Time to reach peak concentration (tmax) and peak plasma concentrations (Cmax) differed significantly, but both parameters showed a more significant variability after SC than after IM administration. Bioavailabilities after extravascular administration were high (>70%). Therefore, given general adverse reactions that were not observed in any ewe and favourable pharmacokinetics, tildipirosin could be effective in treating bacterial infections in sheep.

## 1. Introduction

Pneumonia commonly develops in sheep, often due to the combination of adverse physical and physiological stress with viral and bacterial infections. Usually, *Mannheimia haemolytica*, *Pasteurella multocida,* and *Mycoplasma ovipneumoniae* can be cultured from the upper respiratory tract of healthy sheep. These specific microorganisms are the principal causes of bacterial pneumonia in sheep when simultaneous infection occurs, enhancing their pathogenicity [1,2,3].

Pneumonia is a significant cause of death in sheep flocks in countries where the production is intensive. Annually, it is estimated that pneumonia causes a strong economic impact because of the reduced growth rate, a downgrading of lamb carcasses, and high mortality of the animals. Thus, the administration of antibiotics with penetration to the respiratory system is essential to treat bacterial pneumonia in sheep to reduce morbidity, mortality, and economic losses [4]. However, small ruminants are a relatively less commercialized species compared to cattle and swine, with limited drugs labeled explicitly for ovine use. Therefore, much of the antimicrobial use that occurs in sheep is classified as extra-labelled use with dosage, administration frequency, indications, and times of drug withdrawal, often being extrapolated from information calculated on other species [5,6]. Moreover, subtherapeutic concentrations can lead to increase the risk of appearance of bacterial resistance when veterinary clinicians provide treatments to sheep without knowledge of antibiotic disposition in these species. As a result, the determination of tildipirosin levels in biological fluids is a prerequisite procedure for dose optimization of tildipirosin therapy through pharmacokinetics in veterinary medicine.

Tildipirosin is a 16–membered ring macrolide antimicrobial, exclusively used in veterinary practice that has been approved for parenteral treatment of respiratory illness in cattle and swine [7,8]. Antimicrobial activity of macrolides is attributable to their binding to the 23S ribosomal ribonucleic acid (rRNA) of the 50S ribosomal subunit of bacterial cells, inhibiting bacterial protein synthesis [9]. Macrolides are classified as bacteriostatic drugs in most bacteria. The antimicrobial activity spectrum of tildipirosin includes *Mannheimia haemolytica, Pasteurella multocida*, *Histophilus somni*, *Actinobacillus pleuropneumoniae*, *Bordetella bronchiseptica*, and *Haemophilus parasuis*, which are bacteria correlated with bovine and swine respiratory illness. In fact, in in vitro studies, it was shown that tildipirosin has a bacteriostatic effect against *P. multocida* and *B. bronchiseptica*, and a bactericidal effect for *M. haemolytica, H. somni, A. pleuropneumoniae*, and *H. parasuis* [10,11,12].

The pharmacokinetics of tildipirosin have been reported in pigs [7], cattle [8], mice [12], rabbits [13], and dogs [14]. In these studies, tildipirosin showed long half-lives and high bioavailabilities after extravascular administration. Further, this drug is immediately absorbed and distributed mainly to tissues, with a distinctive macrolide pattern, low concentrations in plasma with multifold higher concentrations found in peripheral tissues [7,8]. However, no information is available on the pharmacokinetics of tildipirosin after parenteral administrations in sheep. Therefore, the objective of this work was to determine the disposition kinetics of tildipirosin after intravenous (IV), intramuscular (IM), and subcutaneous (SC) administration in sheep.

## 2. Materials and Methods

### 2.1. Animals

Six clinically healthy Montesina ewes weighing 67.5 ± 5.3 kg and aged from 3 to 6 years were selected from the Teaching Veterinary Farm at the University of Murcia (Spain). The health status of the animals was determined through physical examination, hematology and, clinical biochemistry (albumin, bilirubin, GGT, AST, creatinine, and urea). For each treatment period, ewes were observed regularly for general physical status, and clinical examinations were recorded before injection and at 2, 12, 24, 48, and 72 h post-injection. Throughout the experiment, the ewes were fed with alfalfa hay, concentrate free of any drug, and provided water ad libitum. No animals were treated with any drug for at least 30 days previous the study. The Bioethical Committee of the University of Murcia (Spain) approved the experimental protocol (CEEA 558/2019) before the study.

### 2.2. Experimental Design

A crossover model (2 × 2 × 2) was designed in three periods, with at least a 30-day washout between periods. Each ewe received a single IV, IM, and SC injection of tildipirosin randomly (Zuprevo 180, MSD Salud Animal, Salamanca, Spain) at a dose of 2 mg/kg (IV administration) or 4 mg/kg (IM and SC administrations). Intravenous administration was directly injected into the left jugular vein. Subcutaneous injection was administered in the thoracolumbar region lateral to the mid-line, and intramuscular injection was applied into the semimembranosus muscle. Blood samples were collected from the right jugular vein into heparinized tubes at 0 (pre-treatment), 0.083, 0.167, 0.25, 0.5, 0.75, 1, 1.5, 2, 4, 6, 8, 10, 12, 24, 32, 48, 72, 96, and 120 h post-treatment, and centrifuged at 600× *g* for 10 min. The plasma was recovered and stored at −40 °C until analyzed.

Damage at administration sites was evaluated by observing lameness, changes in skin temperature, and inflammatory reactions. Moreover, muscular damage and inflammatory response were evaluated by creatin kinase (CK) and haptoglobin (Hp), respectively. Also, cardiotoxicity was evaluated by creatine kinase cardiac isoenzyme (CK–MB) and troponin (Tn). For this propose, additional and independent blood samples were obtained at 0 (pre-treatment), 0.5, 1, 2, 3, and 4 days after tildipirosin administration.

### 2.3. Analytical Methods

Plasma concentrations of tildipirosin were measured using an HPLC (High-performance liquid chromatography) method with an ultraviolet detector. An Agilent series 1220 Infinity LC HPLC system (Agilent Technologies Spain, Madrid, Spain) was equipped with a dual gradient pump, a manual injector, and a variable wavelength detector. The system mentioned above was connected to a Gilson 234 Autoinjector for HPLC systems (Gilson Incorporated, Middleton WI, USA).

Tildipirosin pure substance (used for quality controls) and tylosin tartrate (used as internal standard) were obtained from Cymit Química (Barcelona, Spain). Aliquots of 450 μL sheep plasma were obtained from each sample to which 10 µL of the internal standard solution plus 900 µL of acetonitrile were added. Subsequently, it was homogenized in a vortex for 1 min. Plasma proteins were precipitated by shaking in an ultrasonic bath for 5 min, followed by centrifugation for 10 min at 1200 *g*. The supernatant was extracted (1100 μL) and transferred to other polypropylene tubes and evaporated for 4 h at room temperature (20 °C) in a SpeedVac Vacuum Concentrators (Fisher Scientific, Madrid, Spain). The residue was reconstituted with 75 μL of the mobile phase, and 50 μL was injected into the HPLC system. The HPLC separation was performed using a reverse-phase Zorvax Eclipse XDB-C18 column, 150 × 3.0 mm, 5 μm particle size (Agilent Technologies Spain, Madrid, Spain), and the column temperature was set at 30 °C. The mobile phase was composed of 0.3% formic acid (phase A) and acetonitrile (phase B). The gradient was programmed as follows (minute/A%:B%): 0–2/95:5, 15/70:30, 17/55:45, 18–20/95:5, with a constant flow of 1.0 mL/min. The UV wavelength was established at 289 nm. Tildipirosin and tylosin eluted at approximately 6 and 15 min, respectively.

Creatin kinase and CK-MB were determined by spectrophotometry using an automated chemistry analyzer (AU400 Beckman Coulter Analyzer, Nyon, Switzerland). Troponin concentration was measured by an immunoassay kit (Immulite 1000). Finally, Hp was determined by a commercially colorimetric method (Tridelta Pahase range haptoglobin kit; Tridelta Development Ltd., Maynooth, Ireland).

### 2.4. Method Validation

The HPLC method using an ultraviolet detector was performed according to the Food and Drug Administration [15]. Interfering peaks from endogenous compounds in the blank sheep plasma samples were not observed with the retention times of tildipirosin and internal standard. Quality control samples were prepared from a pool of blank sheep plasma spiked with six concentrations of tildipirosin (100, 300, 1000, 1666, 2500, and 3000 µg/L), and stored at −40 °C until analyzed. Calibrators and quality control samples were extracted as described above and injected into the chromatographic system. The linearity, percentage of recovery, repeatability, reproducibility, lower limit of quantification (LLOQ), and detection limit (LOD) were calculated before starting the experiment.

### 2.5. Pharmacokinetic Analysis

Pharmacokinetic parameters were estimated for each ewe using the WinNonlinTM software package (WinNonlin Professional version 5.1.; Pharsight Corporation, Mountain View, CA, USA). Noncompartmental parameters calculated were: plasma drug concentration immediately after intravenous administration (C_0_), slowest disposition (elimination) rate constant (λ_z_), elimination half-life associated with the terminal slope (λ_z_) of a semilogarithmic concentration-time curve (t_½λz_), area under the plasma concentration-time curve from zero to 24 h (AUC_0-24_), area under the plasma concentration-time curve from zero to infinity (AUC_0–∞_), mean residence time (MRT), mean absorption time (MAT), systemic clearance (Cl), apparent volume of distribution at steady state (V_ss_) and apparent volume of distribution calculated by the area method (V_z_). Peak plasma concentrations (C_max_) and times to reach peak concentration (t_max_) were estimated directly from the experimental data. Bioavailability (F) was calculated by the method of corresponding areas: F (%) = (AUC extravascular/AUC intravenous) x (Dose intravenous/Dose extravascular) × 100. The area under the plasma concentration-time curve extravascular and AUC intravenous were truncated by deconvolution to calculate bioavailability.

### 2.6. Statistical Analysis

Pharmacokinetic data were presented as the geometric mean and standard error of the mean (SEM). Harmonic means were calculated for half-lives of disposition. Pharmacokinetics parameters were compared by Wilcoxon rank-sum test between IV, IM, and SC administration. Comparations between times were performed by Friedman test for cardiotoxicity, muscular damage, and inflammatory response. A significant difference was considered when *p* < 0.05. Statistics analysis was performed with IBM SPSS Statistics 24 (New York, NY, USA).

## 3. Results

### 3.1. Animals

Throughout the experiment, all ewes were healthy, and local or systemic adverse reactions were not observed during or after IV, IM, and SC administration of tildipirosin. Indeed, Hp, CK-MB, and Tn showed no variations after the administration of tildipirosin. However, an increase in CK was determined after the IM administration of tildipirosin, although this increase was short-term (24 h). Mean (± SEM) for Hp, CK-MB, CK, and Tn concentrations in plasma of ewes after IV, IM, and SC administration of tildipirosin are shown in Appendix A.

### 3.2. Analytical Method

The linear regression equation of tildipirosin in plasma was y = 0.0003x − 0.0179, with a regression coefficient values of r = 0.998. The variation coefficient for the repeatability and reproducibility were < −3.7% and < 12.7%, respectively. The recovery of tildipirosin from plasma was 88.9%. Finally, the LLOQ and LOD were 100 μg/L and 75 μg/L, respectively.

### 3.3. Pharmacokinetics

This report is the first description of the pharmacokinetics of tildipirosin after IV, IM, and SC administration in ewes. Plasma concentrations above the LLOQ were determined until the collection time points 32, 96, and 120 h after IV, SC and IM administration, respectively. Mean (± SD) plasma concentrations of tildipirosin after IV, IM, and SC administrations are shown in Figure 1.

The drug-plasma concentration-time profiles of tildipirosin showed a faster absorption and higher peak plasma concentration after IM injection than SC administration. Significant differences (*p* < 0.05) between the two extravascular groups were found in times to reach peak concentration (t_max_) and peak plasma concentrations (C_max_). Mean (± SEM) non-compartmental pharmacokinetic parameters are presented in Table 1.

## 4. Discussion

The essential characteristics of macrolide antibiotics are the presence of high tissue concentrations in tissues and significant accumulation in phagocytic cells that remained for a long time after plasma concentrations declined to low levels. That is even more apparent in the azalide sub-class as azithromycin, gamithromycin, or tulathromycin [16,17,18,19]. High tissue concentrations per se are not usually thought to be relevant to efficacy, especially for extracellular organisms. However, disease studies using animal models have demonstrated a positive correlation efficacy with extravascular or tissue concentrations of the antibiotics mentioned above the minimum inhibitory concentration (MIC) for infecting organisms. At the same time, for macrolides, it has been reported that high tissue concentrations are achieved, although serum/plasma concentrations remained below the MIC [20,21,22], and this has also been found for tildipirosin [23].

Following a single IV injection of tildipirosin to ewes, the apparent volume of distribution was high (5.36 ± 0.57 L/kg), indicating a wide distribution in the body, which can be attributed to a significant fraction of tildipirosin inside the cells or ion-trapping tissues. This value is lower than those reported in pigs, rabbits, cattle, and dogs [8,13,14,23], perhaps due to the analytical technique used in the present study. Nevertheless, all data exceeds 0.6 L/kg, which is the volume corresponding to the body extracellular fluid.

Tildipirosin was slowly cleared from sheep plasma (0.21 L/h/kg), but it was more promptly cleared and had a shorter elimination half-life in sheep than in cattle (0.14 L/h/kg) [8]. However, tildipirosin administrated in dogs was cleared at a faster rate (0.72 L/h/kg) [14], but unexpectedly resulted in a longer half-life than in the present study.

After extravascular administrations, half-lives of tildipirosin increased compared to IV injection due to the enlargement provoked by the absorption phase. The half-life after SC administration appears to be longer than the after IM treatment, but they were not significantly different. Consistently, MRT values followed the same pattern. In the case of MAT after SC administration, it is longer than MRT after IV treatment, indicating that tildipirosin follows a flip-flop model in which the absorption is the limiting step for eliminating plasma. This would elicit considerable variability in drug disposition, and this phenomenon is not evident in the case of IM administration. Half-lives of tildipirosin in sheep seem to be shorter than those reported in other studies and species [7,8,12,14], maybe because the analytical technique reaches lower quantification levels than in the present study. Such concentrations could overestimate this parameter and quantify residues without any or slight antibacterial activity can only contribute to the development of resistant bacteria.

Following IM administration, tildipirosin peak plasma concentration is reached faster (t_max_ = 0.62 ± 0.10 h) than SC administration. Peak plasma concentrations (C_max_) and times to reach peak concentration (t_max_) differed significantly, but both parameters showed a more considerable variability after SC than after IM administration. Peak plasma concentrations after IM administration are similar to those obtained in dogs [14] and pigs [7], but the t_max_ is delayed in the case of sheep. After SC injection, the average C_max_ was higher in sheep (C_max_ = 711 µg/L) than in cattle (C_max_ = 584 µg/L) [8]. The t_max_ showed wide variability (2.67 ± 1.60 h) in our study, but significant variations in the rate of absorption had also been reported after SC administration in sheep for gamithromycin (from 0.16 to 6 h) [17] and tulathromycin (1.6 ± 2.2 h) [16]. This high individual variability could be attributed to differences between animals (fat deposition, hepatic, gastrointestinal, and renal function). Meanwhile, the variable drug absorption rates among individual animals could be attributed to the use of a single site drug administration. Furthermore, because tildipirosin is an extended-release formulation [7], the absorption process could be influenced by the flip-flop kinetics. The AUC in sheep is different from cattle, pigs, and dogs, which could be due to differences between these species. Also, studies conducted in pigs reported different values of AUC, although the same dose and route were used [7,23]. Tildipirosin showed acceptable absolute bioavailabilities following IM and SC administration (79.1% and 68.7%, respectively). Furthermore, high values for absolute bioavailability have been reported in rabbits, pigs, and dogs [8,13,14].

It has been reported that after IM or SC administrations of tildipirosin is rapidly distributed to the respiratory tract of pigs and cattle at levels 681 or 31 times higher in bronchial fluid than in plasma at 120 h after a single drug administration, respectively. It could be assumed that such high concentrations could also be reached in sheep lung tissues as high absolute bioavailability has also been obtained after both routes of administration. Further investigation is required to get detailed information of tildipirosin to confirm this assumption in sheep.

Most macrolides are classified as time-dependent killing drugs and best described by the PK/PD parameter time above MIC (T > MIC). However, for new generation macrolides, such as gamithromycin, tulathromycin, and tildipirosin, AUC_0–24_/MIC is the best surrogate index to correlate with a successful therapeutic effect [12,24]. Zeng et al. (2018) demonstrated bacteriostatic action, a 1–log_10_ kill, and 2–log_10_ kill of bacterial counts effect of unbound tildipirosin against *P. multocida* in a neutropenic murine lung infection model with AUC0–24/MIC values of 19.9, 31.9 and 53.3 h, respectively. The MIC values of tildipirosin have not been reported from bacterial sheep isolates. However, MIC_90_ values of tildipirosin against the pathogens most commonly involved in the etiology of bovine respiratory disease were determined as 1 μg/mL for *M. haemolytica* and *P. multocida* and 4 μg/mL for *H. somni* [10]. In the case of tildipirosine in cattle, it has been shown that concentrations in lungs exceed the MIC_90_ of all target pathogens (1–4 µg ⁄mL) for 16 days (H. somni) or at least 28 days (M. haemolytica and P. multocida), demonstrated in clinical field trials for periods of 21 days, although plasma concentration were under those MIC_90_ levels since the first moments after drug administration [10]. Tildipirosin and other macrolides (tulathromycin, gamithromycin, azithromycin, and tilmicosin) have shown clinical efficacy, although plasma concentrations are lower than the MIC_90_ levels [7,8,21,22,25], but high lung tissue concentrations are reached. Treatment with tildipirosin potentially offers the advantage of less frequent administration over a shorter duration because of its synergism with serum components and intracellular enzymes, increasing antibiotic uptake by phagocytes and efficacy of intracellular bactericidal enzymes, as demonstrated with other azalides [26].

## 5. Conclusions

It is concluded that, considering that general adverse reactions were not observed in any ewes of the study, and the favourable pharmacokinetic properties such as the high apparent volume of distribution, acceptable bioavailabilities, and long half-lives, tildipirosin administered at 4 mg/kg after IM and SC administrations could be effective to treat bacterial infections in sheep. However, further studies are needed to establish concentrations of tildipirosin in lung tissue and clinical efficacy against specific pathogens.

## Figures and Tables

**Figure 1 animals-10-01332-f001:**
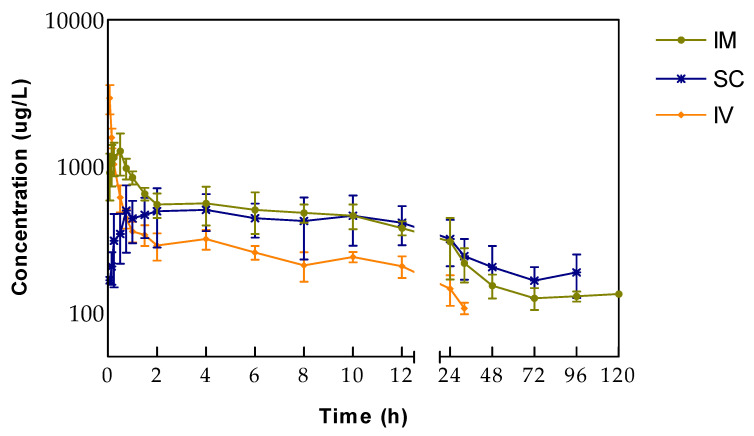
Semilogarithmic plots of tildipirosin concentrations in ewes after intravenous, intramuscular and subcutaneous administration. Values are arithmetic mean ± SD (*n* = 6).

**Table 1 animals-10-01332-t001:** Pharmacokinetic parameters (geometric mean ± SEM) of tildipirosin in ewes (*n* = 6) after intravenous, intramuscular and subcutaneous administration at a single dose of 2, 4 and 4 mg/kg, respectively.

Parameters (Units)	Intravenous	Intramuscular	Subcutaneous
C_0_ (µg/L)	5281.9 ± 799.4		
λ_z_ (h ^−1^)	0.040 ± 0.006	0.024 ± 0.010	0.015 ± 0.002
t_½λz_ (h) *	17.16 ± 2.25	23.90 ± 6.99	43.19 ± 5.17 a
V_Z_ (L/kg)	5.36 ± 0.57		
V_ss_ (L/kg)	4.93 ± 0.58		
Cl (L/h/kg)	0.21 ± 0.01		
AUC_0–24_ (µg·h/L)	6002.0 ± 322.6	10452.4 ± 496.2	9565.5 ± 1010.7
AUC_0–∞_ (µg·h/L)	9395.1 ± 571.8	22,619.6 ± 3590.2	30,558.9 ± 4344.2
MRT (h)	22.68 ± 3.23	39.70 ± 10.07	65.21 ± 7.87 a
MAT (h)		18.73 ± 5.49	38.98 ± 8.95
C_max_ (µg/L)		1264.4 ± 163.4	584.5 ± 75.7 b
t_max_ (h)		0.62 ± 0.10	2.67 ± 1.60 b
F (%)		79.17 ± 4.68	68.78 ± 7.47

***** Harmonic mean, a: Significantly different from IV (*p* < 0.05). b: Significantly different from IM (*p* < 0.05). C_0_: serum drug concentration immediately after intravenous administration; λ_z_: slowest elimination rate constant; t_½λz_: elimination half-life associated with the terminal slope (λ_z_) of a semilogarithmic concentration–time curve; V_z_: apparent volume of distribution calculated by the area method; V_ss_: apparent volume of distribution at steady state; Cl: total body clearance of drug from plasma; AUC_0–24_: area under the plasma concentration-time curve from zero to 24 h; AUC_0–∞_: area under the plasma concentration-time curve from zero to infinity; MRT: mean residence time; MAT: mean absorption time; C_max_: the peak or maximum plasma concentration following extravascular administration; t_max_: time to reach peak or maximum plasma concentration following extravascular administrations; F: the fraction of the administered dose systemically available (bioavailability).

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
