# Peer review of "Pharmacokinetics of Tildipirosin in Ewes after Intravenous, Intramuscular and Subcutaneous Administration"

_animals, 2020, doi:10.3390/ani10081332_

Round 1

Reviewer 1 Report

  1. This is an interesting study with a potential for further development in the future for use of Tildipirosin as a labelled drug in sheep.There are no other studies on pharmacokinetics of the drug Tildipirosin in sheep. Other study reports cardiotoxicity which is not similar. Thus, this study is novel.
  2. I would like to know if the power analysis for the sample size (number of sheep) was conducted prior to the study.
  3. Information on the approval from Animal Ethics Committee or equivalent authority prior to the study would be desirable.
  4. Overall, this study has been presented in a clear and appropriate format. The introduction is relevant and provides sufficient information on background and necessity/importance of the study. Methods and materials are well described as required. The results and discussion are in line with the significance of the study and convincing. It is a well presented and promising research work.   I would like to let you know that information on power analysis for the sample size (number of sheep used for the study) (if this was performed by the authors or not) is my only concern and I would like to know.   

Author Response

Point 1: This is an interesting study with a potential for further development in the future for use of Tildipirosin as a labelled drug in sheep. There are no other studies on pharmacokinetics of the drug Tildipirosin in sheep. Other study reports cardiotoxicity which is not similar. Thus, this study is novel.

Response 1: We are extending the investigation to the determination of tildipirosin in respiratory mucus and MIC determination in the main pathogens that produce pneumonia in sheep.

Point 2: I would like to know if the power analysis for the sample size (number of sheep) was conducted prior to the study.

Response 2: Wagner (1968) indicates that, in pharmacokinetic trials, when plasma concentration values are normally distributed, the coefficients of variation obtained range between 25% and 75%. Establishing that, the number of suitable animals to use in these studies should be between 5 and 20 (We have used 6 sheep for this study). We have obtained values even lower than that 50% in all extractions times. Moreover , the highest values are much lower than the 75% that Wagner (1968) considers normal, so a priori the data obtained are valid. (WAGNER, J.G., 1968: Pharmacokinetics. Am. Rev. Pharmacols., 8: 67-94).

Point 3: Information on the approval from Animal Ethics Committee or equivalent authority prior to the study would be desirable.

Response 3: The experimental protocol (CEEA 558/2019) was approved before the study started.

Point 4: Overall, this study has been presented in a clear and appropriate format. The introduction is relevant and provides sufficient information on background and necessity/importance of the study. Methods and materials are well described as required. The results and discussion are in line with the significance of the study and convincing. It is a well presented and promising research work.   I would like to let you know that information on power analysis for the sample size (number of sheep used for the study) (if this was performed by the authors or not) is my only concern and I would like to know.

Response 4: This point was resolved in answer to point 2.

Reviewer 2 Report

This is a useful contribution to the literature, but is overinterpreted.

Intro: there should be some mention of antibiotic resistance and the role of subtherapeutic concentrations. MICs should probably be included here.

M&M: Why were the im injections made into the semimembranosus m rather than the neck? Why was cardiotoxicity monitored with CK and troponin? The current view on macrolide cardiotoxicity is that they interfere with K channels in the conducting tissue. If the authors have a different view, they should provide references to support it.

Results: fine.

Discussion: A high Vd indicates that the drug has moved out of the plasma but you don't know where it has gone unless you measure tissue concentrations. The tail end of the plasma concentration / time curve looks variable. If you are going to discuss half lives, you should comment on this. Zuprevo 180 is not an extended release formulation. You must discuss antibiotic resistance and the consequences of persistent low concentrations.

Conclusion: You did not compare tildipirosin to other macrolides and you did not measure tissue concentrations. Please restrict conclusions to your results. You cannot make clinical recommendations based on your results. 

Author Response

Point 1: This is a useful contribution to the literature, but is overinterpreted.

Response 1: We are extending the investigation to the determination of tildipirosin in respiratory mucus and MIC determination in the primary pathogens that produce pneumonia in sheep. The conclusion has been amended.

Point 2: Intro: there should be some mention of antibiotic resistance and the role of subtherapeutic concentrations. MICs should probably be included here.

Response 2: We have clarified the role of subtherapeutic concentrations on antimicrobial resistance (lines 65-67). But at the moment MICs of sheep pathogens are not yet available/published and then it is not possible to include them.

Point 3: M&M: Why were the im injections made into the semimembranosus m rather than the neck?

Response 3: Neck is not used in sheep for intramuscular administration.

Point 4. Why was cardiotoxicity monitored with CK and troponin? The current view on macrolide cardiotoxicity is that they interfere with K channels in the conducting tissue. If the authors have a different view, they should provide references to support it.

Response 4. Both markers were used previously in other articles of macrolides to evaluate cardiotoxicity (B Dik, E Bahcivan, H E Faki and K Uney Tildipirosin may cause cardiotoxicity in sheep. Biomedical Research (2017) Volume 28, Issue 19: 8234-8239; Atli O, Ilgin S, Altuntas H, Burukoglu D. Evaluation of azithromycin induced cardiotoxicity in rats. Int J Clin Exp Med. 2015;8(3):3681-3690) and they are widely recognized and applied to evaluate cardiac damage.

Point 5. Results: fine.

Response 5. Thank you!!

Point 6. Discussion: A high Vd indicates that the drug has moved out of the plasma but you don't know where it has gone unless you measure tissue concentrations. The tail end of the plasma concentration/time curve looks variable. If you are going to discuss half-lives, you should comment on this. Zuprevo 180 is not an extended-release formulation. You must discuss antibiotic resistance and the consequences of persistent low concentrations.

Response 6. We have discussed this fact in lines 210-220 and lines 260-265. Zuprevo is not an extended-release formulation and it has been corrected and this paragraph clarified lines 254-256.

Point 7. You must discuss antibiotic resistance and the consequences of persistent low concentrations.

Response. 7. The characteristic of macrolides is that plasma concentrations are low but at tissue locations, where most infections occur, they reach high and sustained concentrations, then we think it is enough discussed in lines 277-280.

Point 8. Conclusion: You did not compare tildipirosin to other macrolides and you did not measure tissue concentrations. Please restrict conclusions to your results. You cannot make clinical recommendations based on your results.

Response 8. The conclusion has been amended.